# Allogeneic Hematopoietic Transplantation for Multiple Myeloma in the New Drugs Era: A Platform to Cure

**DOI:** 10.3390/jcm9113437

**Published:** 2020-10-26

**Authors:** Alberto Mussetti, Maria Queralt Salas, Vittorio Montefusco

**Affiliations:** 1Clinical Hematology Department, Institut Català d’Oncologia-Hospitalet, 089080 Barcelona, Spain; mqsalas@iconcologia.net; 2Institut d’Investigació Biomèdica de Bellvitge (IDIBELL), 08908 Barcelona, Spain; 3Oncohematology Deapartment, ASST Santi Paolo e Carlo, 20142 Milano, Italy; vittorio.montefusco@asst-santipaolocarlo.it

**Keywords:** multiple myeloma, allogeneic transplantation, immunotherapy

## Abstract

Allogeneic hematopoietic cell transplantation (alloHCT) represents a treatment option for multiple myeloma (MM) patients. As shown in several studies, alloHCT is highly effective, but it is hampered by a high toxicity, mainly related to the graft-versus-host disease (GVHD), a complex immunological reaction ascribable to the donor’s immune system. The morbidity and mortality associated with GVHD can weaken the benefits of this procedure. On the other side, the high therapeutic potential of alloHCT is also related to the donor’s immune system, through immunological activity known as the graft-versus-myeloma effect. Clinical research over the past two decades has sought to enhance the favorable part of this balance, along with the reduction in treatment-related toxicity. Frontline alloHCT showed promising results and a potential for a cure in the past. Currently, thanks to the improved results of first-line therapies and the availability of effective second- or third-line salvage therapies, alloHCT is reserved for selected high-risk patients and is considered a clinical option. For donor lymphocyte infusion, bortezomib or lenalidomide have been used as consolidation or maintenance therapies post-transplant—none has become standard of care. For those patients who relapse, the best treatment should be evaluated considering the patient’s clinical status and the previous lines of therapy. The use of newer drugs, such as monoclonal antibodies or other immunotherapies in the post-transplant setting, deserves further investigation. However, acceptable toxicity and a synergic effect with the newer immune system could be hopefully expected.

## 1. Introduction

Multiple myeloma (MM) accounts for 1% of all malignant diseases and 10% of all hematological malignancies [1]. Novel agents have been incorporated in the treatment of MM over the last two decades [2,3]. This has led to an improvement in the duration of disease response and overall survival (OS) for this group of patients [2,3,4].

According to the American Society for Transplantation and Cell therapy (ASTCT) and European Society for Blood and Marrow Transplantation (EBMT) guidelines, the use of high-dose chemotherapy followed by autologous hematopoietic cell transplantation (autoHCT) is the standard of care for transplant-candidate patients with newly diagnosed MM [5,6]. The implementation of novel agents has led to improvements in outcomes after first-line autoHCT [2,3,4,7]. For these reason, Allogeneic hematopoietic cell transplantation (alloHCT) is currently performed in selected patients in relapse or progression after first-line therapy. AlloHCT as consolidation after first-line induction therapy is still indicated as a clinical option in selected patients [5,6].

According to the last reports provided by the Center for International Blood and Marrow Transplant Research (CIBMTR) and EBMT, MM remains the most common indication for autoHCT in the United States and Europe. However, the proportion of MM patients treated with alloHCT is decreasing. A total of 360 patients with plasma cell disorders underwent alloHCT in 2017. In comparison to 2016, this proportion decreased by 17% [8,9,10]. AlloHCT is a treatment with curative potential in MM due the immune-mediated graft-versus-myeloma (GVM) effect [9,10,11,12,13]. Nevertheless, alloHCT is also associated with considerable therapy-related mortality (TRM), impact on quality of life, and disease relapse. The use of reduced intensity conditioning (RIC) regimens, the refinement of graft-versus-host disease (GVHD) prophylaxis, donor selection, and supportive care has improved alloHCT results [14,15,16]. However, with the development of novel therapeutic strategies, the role of alloHCT in MM requires a critical review [17,18,19]. In this review, we summarize the evidence behind alloHCT in MM and we suggest a possible role of newer therapies in this setting.

## 2. Methods and Methods

We searched the PubMed database using the terms “multiple myeloma” and “allogeneic hematopoietic cell transplant”. Additional searches were undertaken to identify articles related to topics relevant to each particular discussion section. All identified articles were read in full, with relevant information extracted and summarized.

## 3. First-Line Allogeneic Hematopoietic Cell Transplantation

The effectiveness of first-line alloHCT in MM has been explored in different prospective and retrospective studies. However, conclusions regarding depth of response, OS, and progression-free survival (PFS) are inconsistent. Additionally, the differences between conditioning regimens, GVHD prophylaxis, patient selection, and duration of follow-up, make the comparisons between trials challenging.

Two prospective trials were conducted by Lokhorst et al. and Barlogie et al. to explore the efficacy of alloHCT in MM using myeloablative preparative regimens [20,21]. The efficacy of alloHCT was compared with autoHCT after induction chemotherapy in patients with de novo MM. Consistent findings regarding high TRM of 40–60% were reported in both studies resulting in a discontinuation of the use of myeloablative regimens in newly diagnosed MM patients.

The implementation of RIC conditioning approaches in the 2000s allowed the expansion of alloHCT by reducing the risk of TRM. Secondary to the reduced toxicity attributed to this transplant modality, alloHCT has been generally performed after autoHCT in a tandem manner in patients newly diagnosed with MM [15,22,23]. Table 1 summarizes the main prospective studies conducted to determine the effectiveness of first-line alloHCT in MM. These trials implemented first-line non-myeloablative alloHCT after induction chemotherapy and autoHCT. The comparative arm included patients treated with induction treatment and tandem autoHCT. Additionally, a so-called “genetic randomization” based on the availability of a matched related donor (MRD) was generally the reason to be assigned to the alloHCT arm.

Bruno et al. prospectively compared outcomes in 245 MM patients treated with induction treatment and first autoHCT followed by MRD alloHCT vs. tandem autoHCT in 2007 [24]. Superior medians for PFS and OS were documented in patients treated with alloHCT (35 months vs. 29 months; *p* = 0.02, 80 months vs. 54 months; *p* = 0.01; respectively). TRM did not differ between both groups (*p* = 0.09). Giaccone et al. published long-term results in 2011 confirming the evidence reported in the previous study [25]. Long term PFS and OS were significantly superior in the alloHCT group. Garban et al. published results in 2006 from a prospective trial including 284 patients with high-risk MM treated with MRD alloHCT vs. tandem autoHCT after induction and first autoHCT. Median PFS and OS did not differ significantly between both groups (19 months vs. 22 months; *p* = 0.07, 24 vs. 48; *p* = 0.58, respectively) [31]. A long-term analysis was conducted by Moreau et al. supporting the conclusions reported in 2006 [32]. Rosiñol et al. published results in 2008 from 110 patients failing to achieve at least near complete remission after induction and first auto-HCT and assigned them to undergo MRD alloHCT vs. tandem autoHCT. A non-significant trend to lower PFS (median not reached vs. 31 months; *p* = 0.08) was attributed to the 25 patients treated with alloHCT. However, a trend toward higher TRM (16% vs. 5%, *p* = 0.07), was seen in the alloHCT group [26]. Consecutively, in 2011, a “genetic randomization” trial including 710 patients with intermediate/high-risk MM patients was published by Krishnan et al. [27]. PFS and OS were comparable between the two groups. A long-term analysis was published in 2020 to better capture the impact of GVM effect by Giralt et al. [33]. Patients treated with autoHCT-alloHCT had a significant durable reduction in risk of relapse, and a better PFS. The EBMT conducted a prospective study comparing outcomes between 357 adults with MM treated with first-line autoHCT-alloHCT vs. autoHCT-autoHCT [28]. Results were published by Björkstrand et al. in 2011, and a long-term analysis was published by Gahrton et al. in 2013 [34]. PFS and OS were superior for patients treated with alloHCT after induction and first autoHCT (eight-year PFS and OS were 22% vs. 12%; *p* = 0.02, and 49% vs. 39%; *p* = 0.03, respectively). TRM was higher in the alloHCT group (three-year TRM was 13% vs. 3%; *p* = 0.0004). Lokhorst et al. prospectively evaluated results on 260 patients treated with tandem autoHCT-alloHCT vs. auto-autoHCT [29]. This study did not show differences in PFS and OS between both arms. TRM was higher in the alloHCT group (six-year TRM 16% vs. 3%; *p* < 0.001). The last trial conducted to compare autoHCT-alloHCT vs. autoHCT-autoHCT was conducted in high-risk MM patients with deletion of chromosome 13q, and it was published by Knop et al. in 2019 [30]. Patients that received alloHCT after autoHCT had higher PFS (median 34.5 vs. 21.8 months; *p* = 0.003), and comparable OS rates (median 70.2 vs. 71.8 months; *p* = 0.856) than patients treated with autoHCT-autoHCT. TRM was higher in the autoHCT-alloHCT group (two-year TRM 14.3% vs. 4.1%; *p* = 0.008).

Retrospective studies from international transplant registers supported the results provided by these prospective trials [35,36]. However, prospective trials were conducted in patients who had not received novel therapy-based induction regimens. Novel agents have been incorporated on the treatment of MM over the last two decades resulting in an improvement in the duration of the disease response and OS for patients with MM. Additionally, results on alloHCT have notably improved in the last two decades. Prospective randomized trials incorporating new therapeutic strategies in combination with alloHCT are needed to determine the role of alloHCT in newly diagnosed MM.

## 4. Allogeneic Hematopoietic Cell Transplantation in the Relapsed Setting

In contrast to the upfront setting, the role of alloHCT for relapsed MM has not been extensively studied. Prospective studies focused on the use of alloHCT in relapsed MM are very few, and there have been no prospective randomized trials comparing alloHCT to autoHCT in the relapsed setting. Additionally, results from comparative studies are limited by their retrospective nature and/or their small study populations. Main studies exploring the role of alloHCT in relapsed MM are summarized in Table 2.

The EBMT conducted a prospective multicenter study including 49 adults with relapsed MM treated with alloHCT after induction treatment and previous autoHCT [37]. The overall response rate was 95%, including 46% complete response (CR). Five-year PFS and OS were 20% and 26%, respectively, and one-year TRM was 25%. A retrospective analysis was performed by the CIBMTR to compare the outcomes of a second autoHCT vs. alloHCT in 2014 [38]. PFS and OS were superior in the autoHCT group in comparison with the alloHCT cohort (three-year OS and RFS were 6% and 20%, and 12% and 46%, respectively). TRM was higher in patients treated with alloHCT (one-year TRM was 13% vs. 2%).

A multicenter intention-to-treat analysis was conducted by Patriarca et al. to explore the efficacy of alloHCT in MM patients who relapsed after autoHCT and were treated with a salvage therapy based on novel agents [39]. Data were analyzed retrospectively. Improved PFS (two year PFS was 42% vs. 18%; *p* = 0.001) and Non-relapse Mortality (NRM) (two-year NRM was 22% vs. 1%; *p* = 0.001) was reported in the alloHCT cohort with comparable OS rates between both groups (two-year OS was 54% vs. 53%; *p* = 0.329). Passera et al. conducted a retrospective analysis to report data from alloHCT using unrelated donors in relapsed MM [35]. The results from alloHCT using myeloablative, reduced-intensity, and non-myeloablative conditioning were compared. Median OS and PFS between the three cohorts were 29 and 10 months, 11 and 6 months, and 32 and 13 months, respectively (*p* = 0.039 and *p* = 0.049). NRM rates were comparable between all groups (*p* = 0.745).

Finally, over recent years the use of haploidentical donors expanded the use of alloHCT also to patients without a human leukocyte antigen (HLA)-identical donor. In the MM setting, a few studies reported outcomes in this setting [40,41]. Castagna et al. showed retrospectively that the use of a haploidentical donor with post-transplant cyclophosphamide (PTCy) is feasible with a promising OS of 63% and an acceptable TRM of 10% at 18 months. In a more recent CIBMTR/EBMT study, these results were confirmed with a 2-year OS of 48% but a dismal 2-year PFS of 17%. Interestingly, the use of PTCy and bone marrow-derived grafts were associated with better OS. The conclusions are that haploidentical donor can be used in this setting with acceptable results.

Patient’s characteristics in the relapsed setting are quite heterogeneous. The majority of patients received between 3 and 6 prior lines of therapy, comprising autoHCT. For this reason, it is not possible to recommend or suggest alloHCT based on the previous lines of therapies received. However, what emerges from the majority of these studies is that an adequate disease burden reduction (≥partial remission) is essential to attain better outcomes in terms of OS and PFS.

In summary, the effectiveness of alloHCT in the salvage setting is supported by a few prospective and retrospective analyses. However, TRM and disease relapse are still important issues when considering alloHCT in the relapse setting. New drugs are available today for the treatment of first-line and relapsed MM, and the potential use of these drugs as a maintenance therapy in selected patients is being explored. The role of alloHCT needs to be explored in combination with modern therapeutic approaches in prospective randomized trials to better integrate this therapy in relapsed MM patients.

## 5. Current Anti-Relapse Strategies after Allogeneic Stem Cell Transplant

Although alloHCT represents a potentially powerful strategy against MM, disease relapse remains the most frequent cause of death after transplantation. In order to reduce the number of relapsed patients after alloHCT or treating relapse post-alloHCT, different therapeutic strategies have been tried as consolidation, maintenance, or most commonly at relapse. The majority of studies in this setting are of the retrospective type. There are no evidence-based recommendations regarding how to treat relapse post-alloHCT and each case deserves specific considerations. Keeping this in mind, we have some data regarding post-alloHCT use of donor lymphocyte infusions (DLIs)—lenalidomide and bortezomib. Initial data regarding monoclonal antibodies have been recently reported while we do not have published data regarding newer drugs such as immunotherapies. A summary of therapeutic strategies post-alloHCT are reported in Table 3.

## 6. Consolidation and Maintenance

Donor lymphocyte infusions have been used as maintenance therapies. In a recent retrospective study by Groger et al., 61 MM patients without disease relapse/progression or GVHD received escalating doses of DLIs [41]. Of note, 61% of patients received alloHCT as frontline therapy. As a result, the 8-year PFS and OS were 43% and 67%, respectively. Fifty-four percent of patients upgraded their disease response. Acute GVHD grade 2–4 was 33% while moderate-severe chronic GVHD was 13%. However, these patients were all without immunosuppression and the median time from alloHCT to DLIs administration was 10 months. Bortezomib maintenance was used by Green et al. in a prospective phase 2 trial [48]. Patients had high-risk disease and received a tandem autoHCT followed by RIC alloHCT from HLA-identical donors. Bortezomib maintenance at a dosage of 1.6 mg/m^2^ intravenous or 2.6 mg/m^2^ subcutaneously every 14 days was started between 60 and 120 days after alloHCT and continued up to 9 months. Four-year PFS and OS were 52% and 61%, respectively. Acute and chronic GVHD incidence was acceptable and 2-year TRM was less than 15%. In a recent abstract from LeBlanc et al., the preliminary results of a phase 2 trial using bortezomib as maintenance therapy were presented [49]. The study design was similar to the previous study. High-risk patients received tandem autoHCT-RIC alloHCT followed this time by bortezomib during induction and as maintenance at 1.3 mg/m^2^ every 2 weeks for 1 year. At 2 years, PFS and OS were 46% and 92%, respectively. Interestingly, the immunophenotipic complete response improved from 28% pre-alloHCT to 61% post-alloHCT. Acute and chronic GVHD incidences were acceptable. Finally, the preliminary results of a prospective study from European Myeloma Network using bortezomib as part of the conditioning regimen and consolidation for RIC alloHCT were reported [50]. In this study, none of the patients received upfront alloHCT and 87% of the study cohort had already received autoHCT. Bortezomib was administered during as part of the conditioning regimen (on days −9 and −2), as part of GVHD prophylaxis with tacrolimus and methotrexate, as consolidation with or without lenalidomide. Bortezomib, having anti-GVHD properties, can be safely used after alloHCT [46]. This makes it potentially safer when compared to DLIs and lenalidomide. Ixazomib, a newer oral proteasome inhibitor, is under investigation as maintenance after alloHCT for high-risk MM in a phase 2 randomized double-blind trial (BMT-CTN1302) [72]. Ixazomib will be administered between 60 and 120 days post-alloHCT. Treatment will continue up to 12 cycles.

Lenalidomide has been used for maintenance post-alloHCT [54,55,56]. However, due to it inducting IL2, an increased rate of acute GVHD has been observed and limits its use in this context. In the HOVON 76 trial, 10mg of lenalidomide for 21 days in 28 days cycle (up to 24 cycles) was administered in a prospective trial (30 patients) [54]. The majority of patients started 3 months after alloHCT (immunosuppressive treatment was not reduced during the first two cycles of lenalidomide). Interestingly, 43% of patients had to stop treatment because of GVHD incidence (acute GVHD > grade 2 in 11 patients, chronic extensive GVHD in five patients). GVHD developed at a median of 18 days since lenalidomide initiation. The conclusion of the authors was against lenalidomide use as maintenance post-alloHCT. In a more recent phase 2 trial from Alsina et al., 30 patients with high-risk MM received lenalidomide as maintenance post-alloHCT [55]. Dosage was the same of the Hovon 76 trial, but the maximum number of cycles was 12. Dose escalation up to 25mg daily was used in case of no toxicity. Lenalidomide treatment started at a median of 96 days after alloHCT. Additionally, in this study, 37% of patients had to stop lenalidomide because of acute GVHD incidence. The same results were reported in a prospective phase 1/II study by Wolschke et al. [56]. Lenalidomide treament started at a median of 135 days post-alloHCT. The majority of patients received 5 mg QD. Again, all grade acute GVHD was 38% representing the main cause for study discontinuation. Due to its toxicity, further studies are required before recommending lenalidomide for maintenance post-alloHCT.

## 7. Therapy at Relapse

Disease relapse is the most common cause of death after alloHCT. Despite the fact that alloHCT could offer immunological protection known as the GVM effect, once the patient relapses there are no standard recommendations. In most cases these patients are excluded from clinical trials and evidence generally comes from retrospective monocentric studies. Therapy at relapse depends on previous treatments and available treatments at the time of relapse. In a recent study by Montefusco et al., 43 patients who relapsed after alloHCT were studied [73]. At 5 years, 35% overall survival was reported. The authors found that age at transplant >55 years, and the presence of plasmacytomas and chemorefractory disease before alloHCT were associated with decreased survival at relapse. Another very recent study from Chhabra et al. reported the survival outcomes of 60 patients who relapsed after alloHCT [74]. Median survival from relapse was 1.8 years. This time, high-risk cytogenetic, relapse <12 months from alloHCT and acute GVHD before relapse were associated with worse survival. Keeping in mind the considerations about the heterogeneity of treatments post-alloHCT, there are a few studies reporting the effects of specific agents in this setting. In the past, donor lymphocyte infusions were possibly the first agent to be used in relapse or progression after alloHCT. In a historical paper by Van de Donk et al., 63 patients with relapsed or persistent disease after RIC alloHCT received DLIs [43]. In the 38% of patients who responded to therapy, median PFS was 27 months and median OS was not reached. For the whole cohort, acute GVHD incidence was 42% and chronic GVHD was 38%. In a more recent phase 2 study, DLIs were used after cytoreduction with bortezomib-dexamethasone (three cycles) in relapsed/progressed MM patients [44]. Sixteen patients received therapy. The response rate was 62% after the three planned bortezomib-dexamethasone cycles and 68% after DLIs. A significant upgrade in responses was observed after DLIs. Progression-free survival and OS at 3 years were 31% and 73%, respectively. Most interestingly, no grade 3–4 acute GVHD or extensive chronic GVHD were observed. This is possibly explained by the previous use of bortezomib and its immune regulatory effect before DLIs. Bortezomib was used as a salvage treatment for 37 patients in a retrospective study by El-Cheikh et al. [51]. Treatment initiation was at 20 months from transplant. The overall response rate was 73% without significant toxicities. However, the median follow-up was only 9 months. Ixazomib and carfilzomib have been used after alloHCT but more data are required [74].

Lenalidomide is probably the drug which was studied the most for relapse post-alloHCT. Its immunomodulatory effects make it a good candidate to elicit a graft-versus-myeloma effect after transplant. Spina et al. showed for the first time in a retrospective manner that lenalidomide with or without dexamethasone was associated with prolonged OS if the drug was administered post-alloHCT [57]. Similar results were reported from the long-term outcomes of the EBMT-NMAM2000 study, where OS was longer for patients who received tandem autoHCT-alloRIC HCT compared to tandem autoHCT-autoHCT (11.4 vs. 3.9 years) [58]. It is speculated that DLI infusions and a stronger effect of lenalidomide in the autoHCT-alloHCT group had an impact on these results. Cosman et al. reported retrospective results of lenalidomide with or without dexamethasone for relapse post-alloHCT [59]. The median time of treatment initiation was 24 months from alloHCT. The overall response rate was very promising at 83%. However, only 10% of patients received lenalidomide pre-alloHCT. Of note, grade 2–4 acute GVHD was 31% lower than the incidence reported in studies were lenalidomide was used as maintenance. This study showed how time from transplant is an important factor to consider when lenalidomide could be used as salvage treatment. Finally, Bensinger et al. described in a more recent cohort the result of a prospective phase 2 study [60]. Eighteen patients received lenalidomide post-alloHCT. Of these, eight had received lenalidomide pre-alloHCT. The overall response rate was 56%. Grade 2–4 acute GVHD was reported in eight patients and chronic GVHD requiring treatment was observed in eight patients. Finally, scattered data were reported for pomalidomide but strong evidence is required in this setting [73,74].

For the monoclonal antibodies class, daratumumab is possibly the drug for which there is the most published data. Klyuchnikov et al. reported preliminary results of 16 patients who received daratumumab as a single agent at relapse post-alloHCT [65]. Nine out of fifteen patients had a response. This is promising considering this current and heavily pretreated cohort of patients (median of three previous treatments pre-daratumumab). No cases of GVHD were reported and treatment was well tolerated. In another report, 11 patients received daratumumab post-alloHCT [66]. Patients had previously received a median of six lines of therapy. Fifty percent of patients had an objective response and no significant response was registered. Again, more data are required in this setting. Use of elotuzumab in association with lenalidomide and dexamethazone has been reported, but more data are needed to assess its effect [63]. So far, no results of newer immunotherapies other than monoclonal antibodies have been published in the setting of post-alloHCT relapse.

## 8. Current Recommendations and Patient Selection

International recommendations regarding the use of alloHCT for MM are not consistent. EBMT guidelines suggest upfront alloHCT as a clinical option for standard-risk patients and as standard of care for high-risk patients whenever a HLA-identical donor is available [5]. Additionally, alloHCT is indicated as a clinical option for relapsed/refractory disease after autoHCT. ASTCT guidelines considered alloHCT as a clinical option only for relapsed/refractory disease or plasma cell leukemia (first-line or relapsed/refractory setting). Finally, in a consensus conference from the EBMT, ASTCT and International Myeloma Working Group, more specific indications were given for alloHCT in the relapsed/refractory setting: patient with early relapse (less than 24 months) after primary therapy that included an autoHCT and/or high-risk features (cytogenetics, extramedullary disease, plasma cell leukemia, or high lactate dehydrogenase) [75]. However, these recommendations do not come from randomized trials and are largely based on experts’ opinions. Moreover, the use of newer drugs at relapse was not considered at that time. Currently, alloHCT as first-line consolidation is not representing a clinical option considering the good results obtained with triplet induction treatment followed by autoHCT and lenalidomide maintenance [76]. The definition of high-risk myeloma is a dynamic concept. Different stratifications have been developed over the years depending on genetic/clinical stratification but also on drug resistance. The revised risk stratification for myeloma identifies a class of patients (stage III) with serum 2-microglobulin level >5.5 mg/L and high-risk cytogenetic (del(17p) and/or t(4;14) and/or t(14;16)) or high LDH level with a poor 5-year PFS and OS of 24% and 40%, respectively [1]. Considering the poor survival, the toxicity of an alloHCT could be accepted in this setting. Ideally, alloHCT should always be performed in the context of clinical trials for MM patients considering the existing low level of clinical evidence. Whenever an alloHCT is performed, reduced-intensity conditions should be preferred considering the good and consistent results available in the literature, the low toxicity and the fact that the majority of these patients have already received myeloablative autologous transplants. Haploidentical donors represent an acceptable option whenever HLA-identical donors are not available [40,77]. Since the chemosensitive disease and disease burden of alloHCT are two important prognostic factors, disease reduction (at least a partial remission) should be pursued before alloHCT. No consolidation/maintenance therapies are currently recommended. In case of disease relapse, no standard recommendations could be made. The choice of therapy should be tailored depending on patient status, previous treatments and time from alloHCT.

## 9. Future Perspectives

While the use of alloHCT for MM is decreasing, it has a curative potential. Following international consensus, it should be reserved for high-risk patients who failed the first line of therapy (including autoHCT). The use of alternative donors (haploidentical followed by PTCy) is expanding alloHCT indications. The use of newer drugs should be exploited to reduce the disease before alloHCT. However, little is known regarding the potential effects of these drugs on allografting. Reducing the toxicity of transplants through the improvement of conditioning regimens should be pursued in the coming years. Future trials should be possibly focused on post-transplant minimal residual disease evaluations to allow early disease treatment. Additionally, a better biologic characterization of post-transplant disease relapse should guide which therapeutic strategy could be the most effective.

## 10. Summary Keypoints

AlloHCT represents a potentially curative option for MM in the new drugs era. AlloHCT can be considered as an immunological platform for subsequent salvage therapies, such as lenalidomide therapy. Its high toxicity is the reason why this procedure is not offered to all patients. Its use should be currently considered in the relapsed setting for those patients who are fit with clinically and biologically high-risk disease features. AlloHCT can be potentially associated with newer drugs and should not be considered as the last therapeutic available option. Chimeric antigen receptor T cells, bispecific antibodies, immunoconjugates or immune modulating monoclonal antibodies could be synergic with a new immune system and their use should be tested in the future. Additionally, it is possible that these newer strategies alone or in combinations will substitute the need of performing alloHCT for this disease. In fact, redirecting the patient’s own immune system against myeloma instead of balancing the effects of a graft-versus-tumor and graft-versus-host diseases seems to be a safer and more predictable therapeutic intervention. 

## Figures and Tables

**Table 1 jcm-09-03437-t001:** Main prospective studies conducted to evaluate the effectiveness of first-line allogeneic hematopoietic cell transplantation in multiple myeloma patients.

Main Prospective Studies Conducted to Evaluate the Effectiveness of First-Line alloHCT in MM Patients.								
Reference	Timing	N Total Number of Patients	Study Design	Conditioning	OS	RFS	NRM	Conclusion
Bruno et al., 2007 [24].	1994–2004	245	Intermediate and high-risk MM patients Post-induction and autoHCT followed by alloHCT vs. autoHCT based on MRD availability	TBI 2 Gy vs. MEL 100–200 mg/m^2^	Median 80 vs. 54 months (*p* = 0.01)	Median 35 vs. 29 months (*p* = 0.02)	2y NRM 6% vs 1% (*p* = 0.09)	PFS and OS were superior in patients undergoing alloHCT. TRM did not differ between both groups. Long-term analysis was published by Giaccone et al. in 2001 confirming results published in 2007.
		58						
Garban et al., 2006 [25].	2000–2004	284	Intermediate and high-risk MM patients Post-induction and autoHCT followed by alloHCT vs. autoHCT based on MRD availability	Flu-Bu-ATG vs. MEL 220 mg/m^2^ +/− antiIL-6	Median 34 vs. 48 months (*p* = 0.07)	Median 19 vs. 22 months (*p* = 0.58)	11%	No benefit to alloHCT Long-term results published by Mureau et al. in 2008 supporting results published in 2006.
		65						
Rosiñol et al., 2008 [26].	1999–2004	110	Patients failing to achieve near CR Post-induction and autoHCT followed by alloHCT vs. autoHCT based on MRD availability	Flu-MEL vs. MEL 200 mg/m^2^	Median NR vs. 58 months (*p* = 0.9)	Median 20 vs. 26 months (*p* = 0.4)	NRM 16% vs. 5% (*p* = 0.07)	Higher CR rate after allotransplant but no survival benefit
		25						
Krishnan et al., 2011 [27].	2003–2007	710	Intermediate/high-risk MM patients Post-induction and autoHCT followed by alloHCT vs. autoHCT based on MRD availability	TBI 2 Gy vs. MEL 200 mg/m^2^	3 years OS 77% vs. 80% (*p* = 0.191)	3 yr PFS 43% vs. 46% (*p* = 0.671)	3years NRM 11% vs. 4%(*p* < 0.001)	No benefit to allotransplant in this study Long-term results published by Giralt et al. in 2020 showed a significant durable reduction in risk of relapse and better 6-year PFS for alloHCT patients.
		226						
Björkstrand et al., 2011 [28].	2001–2005	357	Post-induction and autoHCT followed by alloHCT vs. autoHCT based on MRD availability	Flu–TBI 2 Gy vs. MEL 200 mg/m^2^	8years OS 49% vs. 39% (*p* = 0.03)	8years PFS 22% vs. 12% (*p* = 0.02)	13% vs. 3% (*p* = 0.02)	Allotransplant correlated with lower risk of relapse and improved PFS. Long-term analysis was conducted and published by Gahrton et al. in 2013.
		108						
Lokhorst et al. [29].	2003–2005	260	Post-induction and autoHCT followed by alloHCT vs. autoHCT based on MRD availability	TBI 2 Gy vs. MEL 200 mg/m^2^	6years 55% vs. 55% (*p* = 0.19)	6years 28% vs. 22% (*p* = 0.68)	6years NRM 16% vs. 3% (*p* < 0.01)	No benefit to having a related donor but allotransplant was by center preference. Relapse lower for those with donors
		122						
Knop et al., 2019 [30].	2001–2007	381	High-risk MM with deletion of del13q Post-induction and autoHCT followed by alloHCT vs. autoHCT based on MRD availability	Flu-MEL vs. MEL 200 mg/m^2^	Median 70.2 vs. 71.8 months (*p* = 0.856)	Median 34.5 vs. 21.8 months (*p* = 0.003)	2years 14.3% vs. 4.1%; (*p* = 0.008)	Largest trial in high-risk patients and with unrelated donors. PFS and OS was superior in patients treated with alloHCT
		135						

OS = Overall Survival; PFS = Progression-Free survival; NRM = Non-relapse Mortality; MM = Multiple Myeloma; AutoHCT = Autologous Hematopoietic Cell Transplantation; AlloHCT = Allogeneic Hematopoietic Cell Transplantation; MRD = Matched Related Donor; TBI = Total Body Irradiation; MEL = Melphalan; ATG = Anti-Thymocyte Globulin; CR = Complete Response.

**Table 2 jcm-09-03437-t002:** Main studies related to the use of allogeneic hematopoietic cell transplantation in the relapsed/refractory setting.

Reference	Timing	N Total n Allo	Study Design	Conditioning	OS	PFS	TRM	Conclusion
Kroger et al., 2002 [37].	2000–2002	21	Single-arm prospective study AlloHCT using unrelated donors after in relapsed MM patients	Flu-MEL-ATG.	2 years OS 74%	2 years PFS 53%	1 year TRM 26%	Feasibly or alloHCT using unrelated donors in MM patients
Freytes et al., 2014 [38].	1995–2008	289	Retrospective comparative analysis AutoHCT vs. alloHCT in relapsed MM patients after induction treatment	RIC alloHCT Multiple conditioning regimens	3 years OS 46% vs. 12% (*p* < 0.001)	3 years PFS 20% vs. 6% (*p* = 0.038)	1 year TRM 2% vs. 13%, (*p* = 0.07)	AlloHCT was associated with higher TRM and lower survival than autoHCT TRM was higher in patients undergoing alloHCT
		152						
Patriarca et al., 2012 [39].	2002–2008	169	Single-arm retrospective descriptive analysis AlloHCT after relapse in patients prior treated with autoHCT	RIC alloHCT Multiple conditioning regimens	2 years OS 53% vs. 54% (*p* = 0.329)	2 years PFS 18% vs. 42% (*p* < 0.001)	2 years TRM 22% vs. 1%, (*p* = 0.07)	PFS benefit of salvage treatment with novel drugs followed by RIC alloHCT in relapsed MM TRM was higher in patients undergoing alloHCT
		169						
Passera et al., 2013 [35].	2000–2009	196	Single-arm retrospective analysis. AlloHCT using unrelated donors in relapsed MM Study from the Italian Bone Marrow Donor Registry	RIC alloHCT Multiple conditioning regimens	3 years OS 40%	3 years PFS 22%	5 years TRM 33.2%	TRM was comparable between alloHCT using different intensity of preparative regimens
		196						

OS = Overall Survival; PFS = Progression-Free survival; TRM = Toxicity Related Mortality; AlloHCT = Allogeneic Hematopoietic Cell Transplantation; MM = Multiple Myeloma; Flu = Fludarabine; MEL = Melphalan; ATG = Anti-Thymocyte Globulin; MM = Multiple Myeloma; AutoHCT = Autologous Hematopoietic Cell Transplantation; RIC = Reduced Intensity Conditioning; AlloHCT = Allogeneic Hematopoietic Cell Transplantation.

**Table 3 jcm-09-03437-t003:** Drugs available for treating multiple myeloma and their principal mechanism of action in the post-allogeneic transplantation setting.

Drugs	Anti-Myeloma Mechanism of Action	Possible Immunological Synergy after alloHCT	Potential Clinical Use
Donor lymphocyte infusion	DLIs are lymphocytes with polyclonal TCR repertoire. Donor T cells can recognize foreign antigens and different HLA molecules on the recipient tumor and non-tumor cells [42].	DLIs can be used to boost donor immune system after transplant. An increased GVHD is an expected risk of this therapy.	Maintenance/consolidation: acute GVHD incidence of 33% [41]. Relapse: responses between 30–60%, but GVHD incidence is a concern [43,44].
Bortezomib	Induces proteasome 20S inhibition with increased cellular levels of proapoptotic proteins. Additionally, it induces G_2_-M phase cell cycle arrest and apoptosis [45].	Impedes degradation of IkB-alpha and its dissociation from NF-kB, blocking NF-kB activation at lymphocytes and dendritic cell level [46,47]. Anti-GVHD effect with possible reduction in alloHCT related mortality.	Maintenance/consolidation: acceptable rate of GVHD [48,49,50]. Relapse Response rate up to 70%, but limited follow-up [44,51].
Ixazomib	Same as bortezomib, but oral route of administration [52].	Same as bortezomib, but oral route of administration [52].	Anti-GVHD effect, maintenance treatment after alloHCT for high-risk MM high-risk patients. Ongoing phase 2 trial [26].
Lenalidomide	Different mechanisms: (1) degradation of IKZF1 and IKZF3 which are essential for B-cell differentiation and MM cells survival; (2) increased IL2 transcription related to IKZF3 (IL2 transcriptional repressor). This could favor proliferation of NK, NKT and CD4+ T cells [53].	Immunomodulatory properties of lenalidomide could enhance graft-versus-tumor effect. However, at high doses, it could also increase GVHD incidence [54].	Maintenance/consolidation: acute GVHD induction in 30–40% of cases leading to study discontinuation [54,55,56]. Relapse: response rate between 50–80%, time from alloHCT related to GVHD incidence [57,58,59,60].
Elotuzumab	Humanized IgG1 anti-SLAMF7 monoclonal antibody. Dual mechanism of action: (1) activation of NK cells and increased granzyme B release, (2) antibody-dependent cellular cytotoxicity. Furthermore, additional SLAMF7 positive immune system cells (CD8+ T cells, monocytes, dendritic cells) are expected to be activated in a antitumor sense [61,62].	Having a new immune system which has not been inhibited or exhausted from myeloma, could increase the efficacy of elotuzumab in the post-alloHCT setting. NK cells are expected to recover in the first months post-transplant, and their antitumor effect has already been reported for other hematological malignancies in this setting. Theoretical increased GVHD risk	Case reports in association with lenalidomide/dexamethasone [63].
Daratumumab and other CD38 monoclonal antibodies	Direct antitumor effect through Fc-dependent immune effector mechanisms (complement-dependent cytotoxicity, antibody-dependent cellular cytotoxicity, antibody-dependent cellular phagocytosis. Indirect antitumor effect through elimination of CD38 expressing immune system cells (Treg cells, Breg cells, myeloid-derived suppressor cells) [64].	A new marrow microenvironment could favor the effect of daratumumab having less anti-apoptotic molecules such as survinin. This mechanism has been described as a resistance mechanism for daratumumab. Theoretical increased GVHD risk [64].	Relapse: promising responses (around 50–60%) and acceptable toxicity, but preliminary data [65,66].
Chimeric Antigen Receptor T cells	CAR T cells directed against tumor antigen (e.g., B cell maturation antigen, BCMA) enable tumor-killing by means of MHC-unrestricted effect. This is mediated by the binding of a single-chain variable fragment to the tumor target antigen. Since the chimeric receptor contained a costimulation molecule, no other binding s are necessary to activate their effector function [67].	Activation of CAR T cells is independent from MHC complex and costimulatory molecules. This could be beneficial in the context of an immunesuppressed environment such as the post-transplant setting. However, the concomitant use of immunesuppressors could limit the use of CAR T cells in the early post-transplant period. T-cell depletion strategies which reduce the use of post-transplant immunesuprpessors could be useful when planning post-transplant CART. GVHD risk is low [68].	No current clinical trials are investigating this strategy
Bispecific antibodies	Classical bispecific antibodies in oncology are composed by one antigen binding site against CD3 receptor (which activates T lymphocytes), and the other binds monovalently or bivalently to tumor antigens (e.g., BCMA). The union of T lymphocytes with tumor, favors the cytotoxic effect of T cells with subsequent tumor cell destruction [69].	The advantage of this class of drugs is their off-the shelf use. Contrary to CAR T, they rely on an intact immune system. This could represent an issue in the post-transplant setting where CD8+ T cells are expected to recover after 2-8 months and CD4+ T cells after 4–12 months [70].	No current clinical trials are investigating this strategy
Immunoconjugates	Immunoconjugates are constituted by 3 components. (1) a monoclonal antibody which binds to a target antigen (e.g., BCMA); (2) an effector molecule with cytotoxic effect (e.g., mafodotin); (3) a “linker” molecule which release the effector molecule to the cancer cell and not to off-target sites [71].	The antitumor effect relies mostly on the cytotoxic effector molecule, and not to monoclonal antibodies related cytotoxic effects. This is an advantage in the post-transplant setting where the immune system is generally suppressed for the first months.	No current clinical trials are investigating this strategy

DLI = Donor Lymphocyte Infusion; GVHD = Graft-Versus-Host Disease; IkB = Inhibitor of kB; NF-kB = Nuclear Factor-kB; MM = Multiple Myeloma; IkZF = Ikarus Zinc Finger; CAR = Chimeric Antigen Receptor; MHC Major Histocompatibility Complex; BCMA = B Cell Maturation Antige.

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
