# Peer review of "Allogeneic Hematopoietic Transplantation for Multiple Myeloma in the New Drugs Era: A Platform to Cure"

_jcm, 2020, doi:10.3390/jcm9113437_

Round 1

Reviewer 1 Report

A nice summary of the studies done, I would add a clear direction forward, such as

When allo should be considered, what are the clinical trial design suggested, just a prospective rather than summary

Author Response

A perspective regarding future clinical trials focuses and patient selection was reported in the section “future perspectives” as required from the reviewer.

Reviewer 2 Report

It is a very well written manuscript. Very clear and describing the literature very well. With the upcoming results of new T cell modalities, this topic is very timely and relevant to discuss. Because of high relapse rates and toxicity after allo SCT in MM, this is no standard treatment for MM, as also described in this review. However there seems to be a GvM effect and might even be a curative potential. Therefore the potential promising use of immunotherapy/newer T cell modalities might be more emphasized in the manuscript.

Some minor remarks:

Page 7, r299: mieloma = myeloma

Page 7, r304: Whenever an alloHCT is performed, reduced-intensity conditionings should always be used.  is there strong evidence here to really state this? Also consolidation/maintenance seems important in different series, but here the statement about this topic is with more caution. Also the statement for RIC conditioning schemes hould be made with more caution

Page 7, r306: Since chemosensistive disease  chemosensitive

Page 7: Chimeric antigen receptor T cells, bispecific antibodies, immunoconjugates or immune modulating monoclonal antibodies could be synergic with a new immune system and their use should be tested in the future.  this is true. However what is missing in the discussion is that these newer T cell directed modalities might possibly replace alloSCT. While allo SCT show rather good survival curves, and only maintenanace may bring responses back post allo SCT, pointing at GvM effects, this might enhance the promiss for other T cel modalities in MM. Please discuss.

Author Response

Page 7, r299: mieloma = myeloma; CORRECTED

Page 7, r304: Whenever an alloHCT is performed, reduced-intensity conditionings should always be used.  is there strong evidence here to really state this? Also consolidation/maintenance seems important in different series, but here the statement about this topic is with more caution. Also the statement for RIC conditioning schemes hould be made with more caution;

THE SENTENCE WAS CORRECTED IN THE FOLLOWING WAY "Whenever an alloHCT is performed, reduced-intensity conditionings should be preferred considering the good and consistent results available in literature, the low toxicity and the fact that the majority of these patients have already received myeloablative autologous transplant." 

Page 7, r306: Since chemosensistive disease  chemosensitive;  CORRECTED

Page 7: Chimeric antigen receptor T cells, bispecific antibodies, immunoconjugates or immune modulating monoclonal antibodies could be synergic with a new immune system and their use should be tested in the future.  this is true. However what is missing in the discussion is that these newer T cell directed modalities might possibly replace alloSCT. While allo SCT show rather good survival curves, and only maintenanace may bring responses back post allo SCT, pointing at GvM effects, this might enhance the promiss for other T cel modalities in MM. Please discuss.

WE AGREE WITH THE REVIEWER REGARDING A POSSIBLE FAVORABLE EVOLUTION OF IMMUNOTHERAPIES. WE ADD THE FOLLOWING PART TO THE FINAL DISCUSSION PART IN ORDER TO FAVOR A CRITICAL REVIEW OF THE TOPIC."Also, it is possible that these newer strategies alone or in combinations will substitute the need of performing alloHCT for this disease. In fact, redirecting the patient’s own immune system against myeloma instead of balancing the effects of a graft-versus-tumor and graft-versus-host disease seems to be a safer and more predictable therapeutic intervention." 

Reviewer 3 Report

This is a fairly complete review of the role of allografting in multiple myeloma.

Comments:

1.  In the introduction, would define what is meant by first-line allo HCT; after no prior therapy or after an induction therapy ?

2.  The last paragraph in introduction shows that allografting is less utilized in MM than in the past.  The authors could point out why and could also elaborate on this in the discussion; for example, how will availability of CAR-Ts, BiTEs, and other new modalities continue to affect this utilization?

3. In the relapsed setting, how many prior therapies have these patients usually had , and is there a limit after which allografting is futile?  Is there a degree of disease response that should be attained before allografting has benefit?

4 . Table 3 is a nice summary of drugs that could be used after allografting. It might be interesting to comment on how some of these therapies might impact allografting when given prior to SCT or to cite any data regarding this; sparse though it may be.

Minor:

Line 32--on should e In the treatment of MM;

Author Response

1.  In the introduction, would define what is meant by first-line allo HCT; after no prior therapy or after an induction therapy ?

We corrected the sentence in the following way: "AlloHCT as consolidation after first-line induction therapy is still indicated as a clinical option in selected patients"

2.  The last paragraph in introduction shows that allografting is less utilized in MM than in the past.  The authors could point out why and could also elaborate on this in the discussion; for example, how will availability of CAR-Ts, BiTEs, and other new modalities continue to affect this utilization?

As suggested also by reviewer number #2, we discussed this issue in the "Future perspective" part. However, considering the absence of evidence related to the topic, this part was left only as a perspective. The following sentence was added: "Also, it is possible that these newer strategies alone or in combinations will substitute the need of performing alloHCT for this disease. In fact, redirecting the patient’s own immune system against myeloma instead of balancing the effects of a graft-versus-tumor and graft-versus-host disease seems to be a safer and more predictable therapeutic intervention."

3. In the relapsed setting, how many prior therapies have these patients usually had , and is there a limit after which allografting is futile?  Is there a degree of disease response that should be attained before allografting has benefit?
We searched for this data in literature. However, study population characteristics are quite heterogeneous and a final recommendation can not be made. However, we specify this issue adding the following sentence (line 154-158) : "Patient’s characteristics in the relapsed setting are quite heterogeneous. The majority of patients received between 3 to 6 prior lines of therapy, comprising autoHCT. For this reasons, it is not possible to recommend or not suggest alloHCT based on the previous lines of therapies received. However, what emerges from the majority of these studies, is that an adequate disease burden reduction (> partial remission) is essential to attain better outcomes in terms of OS and PFS"

4 . Table 3 is a nice summary of drugs that could be used after allografting. It might be interesting to comment on how some of these therapies might impact allografting when given prior to SCT or cite any data regarding this; sparse though it may be.

We searched for evidence related to newer therapies and the potential effects on allografting. However, considering the absence of studies focusing on the use of newer drugs before alloHCT, it was not possible to find any reference. For this reason, we considered appropriate to add the following sentence (lines 324-325):"The use of newer drugs should be exploited to reduce the disease before alloHCT. However, little is known regarding a the potential effects of these drugs on allografting.  "

Minor:

Line 32--on should e In the treatment of MM;
This was corrected as requested